# The Application of MALDI-TOF MS for a Variability Study of *Paenibacillus larvae*

**DOI:** 10.3390/vetsci9100521

**Published:** 2022-09-23

**Authors:** Anna Kopcakova, Slavomira Salamunova, Peter Javorsky, Rastislav Sabo, Jaroslav Legath, Silvia Ivorova, Maria Piknova, Peter Pristas

**Affiliations:** 1Centre of Biosciences of the Slovak Academy of Sciences, Institute of Animal Physiology, Soltesovej 4-6, 040 01 Kosice, Slovakia; 2Department of Epizootiology, Parasitology and Protection of One Health, University of Veterinary Medicine and Pharmacy in Kosice, Komenskeho 73, 041 81 Kosice, Slovakia; 3Department of Pharmacology and Toxicology, University of Veterinary Medicine and Pharmacy in Kosice, Komenskeho 73, 041 81 Kosice, Slovakia; 4Institute of Biology and Ecology, Faculty of Natural Science, Pavol Jozef Safarik University, Srobarova 2, 041 54 Kosice, Slovakia

**Keywords:** honey bee, *Paenibacillus larvae*, variability, MALDI-TOF, biotyping

## Abstract

**Simple Summary:**

An understanding of *Paenibacillus larvae*, the etiological agent of American foulbrood disease, species diversity is crucial for disease epidemiology investigations. Our data indicate that the protein fingerprinting-based MALDI-TOF method provides a much more thorough insight into *P. larvae* diversity compared to the DNA fingerprinting methods used at present.

**Abstract:**

In recent decades, the significant deterioration of the health status of honey bees has been observed throughout the world. One of the most severe factors affecting the health of bee colonies worldwide is American foulbrood disease. This devastating disease, with no known cure, is caused by the Gram-positive spore-forming bacteria of *Paenibacillus larvae* species. At present, DNA-based methods are being used for *P. larvae* identification and typing. In our study, we compare two of the most advanced DNA-based technologies (rep-PCR and 16S rRNA analyses) with MALDI-TOF MS fingerprinting to evaluate *P. larvae* variability in Central Europe. While 16S rRNA analysis presents a very limited variation among the strains, MALDI-TOF MS is observed to be more efficient at differentiating *P. larvae*. Remarkably, no clear correlation is observed between whole-genome rep-PCR fingerprinting and MALDI-TOF MS-based typing. Our data indicate that MALDI-TOF protein profiling provides accurate and cost-effective methods for the rapid identification of *P. larvae* strains and provides novel perspectives on strain diversity compared to conventional DNA-based genotyping approaches. The current study provides a good foundation for future studies.

## 1. Introduction

The honey bee *Apis mellifera* is an economically important insect species that plays a key role in agriculture as a pollinator of crops and fruits. Honey bees are hosts for a diverse community of microorganisms, including bacteria, fungi, protists, and viruses, some of them being pathogenic. One of the most damaging threats with a lethal impact on bee colonies is the spore-forming Gram-positive bacterium *Paenibacillus larvae*, the causal agent of American foulbrood (AFB).

The disease affects the larval stages of honey bees, and spores are the only infectious form of *P. larvae*. Millions of spores can be produced following the infection and death of one larva. In general, spores are resistant to antibiotics, disinfection, and are extremely long-lived, remaining viable for decades. Although adult honey bees are not killed by AFB, they act as vectors of transmission and spread the disease from infected colonies to healthy ones. At present, the only effective method used for eliminating this highly contagious and destructive disease worldwide is burning the diseased hives and equipment, which causes considerable economic losses [1]. Due to its considerable effect on the economy, AFB has become one of the best-studied honey bee diseases.

AFB outbreaks are most often caused by clonally related bacteria that share inherited biochemical and genetic features, but there remains sufficient diversity at the species level leading to variations in the virulence of *P. larvae*. An improved understanding of microbial biodiversity below the species level is necessary for the proper evaluation of pathogenesis, transmission, recognizing particularly virulent strains, and, eventually, for disease prevention [2].

In recent years, traditional approaches used for *P. larvae* subtyping have been replaced by genetic fingerprinting methods that provide highly sensitive, specific, and useful tools for the diagnosis of AFB. Among them, the PCR amplification of repetitive DNA elements (rep-PCR) has been shown to be the most valuable method. PCR fingerprinting methods reveal considerable diversity among *P. larvae* species and several different genotypes have been described [3,4,5,6,7]. The most widely used genotyping method to differentiate between *P. larvae* strains is PCR using ERIC primers. As described in other studies, at least four ERIC genotypes with varying virulence exist [8], ERIC I and ERIC II being the two most important ones [1]. Only recently, the new genotype V was detected among *P. larvae* isolates obtained from Spain [9].

Despite the unquestionable relevance of DNA-based fingerprinting methods that are used for *P. larvae* genotyping in routine diagnostics, they remain laborious and time-consuming. Therefore, it is essential to develop rapid detection and diagnostic methods that are capable of differentiating between related bacterial isolates of *P. larvae*, thus helping us to track the spread and locate the source of disease epidemics.

In recent years, Matrix-Assisted Laser Desorption/Ionization Time-Of-Flight Mass Spectrometry (MALDI-TOF MS) has revolutionized routine biological sample identification in clinical diagnostic microbiology. In comparison to conventional identification methods that rely on biochemical tests and require prolonged cultivation procedures, MALDI-TOF is a rapid, high-throughput, low-cost, and efficient system that identifies microorganisms directly obtained from colonies in a few minutes [10]. In addition to the identification of microbes at the species level, the technology has also emerged as a powerful tool in bacterial typing with practical applications in epidemiological studies and outbreak control. Type-specific protein peaks in mass spectra can be recognized as markers to identify particular genotypes and subtypes [11]. However, despite the benefits of MALDI-TOF MS for the typing of many human pathogens, the data concerning its performance for veterinary bacterial isolates are relatively scarce.

The aim of the current study is to compare the performance of MALDI-TOF MS and molecular fingerprinting methods for the discrimination of *P. larvae* genotypes. For this purpose, we analyze the genetic variability of *P. larvae* strains isolated from AFB outbreaks in Slovakia and obtained from the international collection of microorganisms isolated in the Czech Republic.

## 2. Materials and Methods

### 2.1. Bacterial Strains and Growth Conditions

Four *P. larvae* strains (CCM 38, CCM 4484, CCM 4488, and CCM 5680) were obtained from the Czech collection of microorganisms in Brno and isolated from the Czech territory. Another five *P. larvae* strains were field isolates collected from infected honeycombs: two of them (D3 and D4) were collected from the village Velka Lodina (48°51′23.7512785″ N, 21°9′18.3509445″ E) and three strains (5S, 10S, and M1) were collected from Kosice (48°44′40.5148953″ N, 21°15′30.4527283″ E) in East Slovakia. Dead larvae were collected from a honeycomb using a wooden pestle and homogenized in 1 mL NaCl 0.9% (*w*/*v*). Aliquots of this biomass were inoculated on an MYPGP agar plate containing 1.5% agar, 1.5% yeast extract, 0.3% K_2_PO_4_, 0.2% glucose, and 0.1% sodium pyruvate, as well as 1.0% Mueller–Hinton broth [12]. All *P. larvae* strains were cultivated on an MYPGP agar plate at 37 °C for 48 h. For the DNA isolation procedure, the strains were cultivated in liquid MYPGP broth at 37 °C for 48 h with vigorous shaking.

### 2.2. MALDI-TOF Identification of Bacteria

For the primary identification of the isolated strains, MALDI-TOF MS (Matrix-Assisted Laser Desorption Ionization Time-Of-Flight Mass Spectrometry) (Bruker Daltonics GmbH, Bremen, Germany) analysis was used. A small quantity of a single, freshly grown overnight colony was applied directly onto a polished steel MALDI target plate as a thin film. Alternatively, the biological material (one bacterial colony) was resuspended in 300 μL of distilled water. Subsequently, 900 μL of absolute ethanol was added, the mixture was centrifuged at 13,000 g for 2 min, and the supernatant was discarded. Thirty microliters of formic acid (70% *v*/*v*) were then added to the pellet and thoroughly mixed by pipetting prior to the addition of 30 μL of acetonitrile to the mixture. The mixture was centrifuged again at 13,000 g for 2 min. One microliter of the supernatant was then placed onto a spot of steel target plate and air-dried at room temperature. Both the microbial film and supernatant of the extracted proteins were overlaid with 1 μL of matrix solution (a saturated solution of α-cyano-4-hydroxycinnamic acid in organic solvent (50% acetonitrile and 2.5% trifluoroacetic acid) and air-dried [13]. MALDI-TOF was performed using a Microflex LT instrument (Bruker Daltonics GmbH, Leipzig, Germany) using FlexControl software (version 3.0 Bruker Daltonics GmbH, Bremen, Germany). The spectra were recorded in the positive linear mode. For each spectrum, 240 shots in 40 shot steps from different positions of the target spot (in automatic mode) were collected and analyzed. The raw spectra obtained from each isolate were imported into Biotyper software version 3.0 (Bruker Daltonics GmbH, Leipzig, Germany, database version 3.3.1.0) and analyzed by standard pattern matching with default settings without any user intervention.

### 2.3. DNA Isolation, PCR Amplification, and the Phylogenetic Analysis of P. larvae Isolates

Genomic DNA was isolated from 5 mL of bacterial culture incubated in MYPGP liquid medium. DNA extraction was performed using the extraction protocol described by Pospiech and Neumann [14]. The quality and quantity of the extracted DNA were verified by electrophoresis in 0.8–1.5% agarose gels in a TAE buffer. DNA fragments were visualized using the Carestream Gel Logic 212 Pro (Carestream Health, Rochester, NY, USA) documentation system.

All polymerase chain reactions (PCRs) were performed in a C1000TM Thermal Cycler (Bio-Rad Laboratories, Richmond, VA, USA). For the genetic fingerprinting procedure, ERIC-PCR analysis was performed using primer pair ERIC1R (5′-ATGTAAGCTCCTGGGGATTCAC-3′) and ERIC2 (5′-AAGTAAGTGACTGGGGTGAGCG-3′) under the conditions specified by Versalovic et al. [15]. To infer the phylogenetic affiliation of the bacterial isolates, the 16S rRNA gene was amplified with universal eubacterial primers fD1 and rP2, according to the conditions described by Weisburg et al. [16].

Amplified 16S rRNA gene fragments were purified using Wizard SV Gel and a PCR Clean-Up purification kit (Promega), according to the manufacturer’s instructions. The purified PCR products of 16S rRNA were ligated into pTZ57R/T plasmid (Fermentas, Germany), and *Escherichia coli* K12 ER2267 (F’ proA+B+ lacIq Δ(lacZ)M15 zzf:mini-Tn10 (KanR)/Δ(argF-lacZ)U169 glnV44 e14-(McrA-) rfbD1? recA1 relA1? endA1 spoT1? thi-1 Δ(mcrC-mrr)114::IS10)-competent cells were transformed with the ligation mixture. The transformation was conducted using the calcium chloride procedure, as described by Maniatis et al. [17]. Recombinant plasmids were isolated with a GenElute Plasmid Miniprep kit (Sigma-Aldrich, St. Louis, MO, USA), and the inserted DNA fragments were sequenced by the dideoxy chain-termination sequencing method at Eurofins Genomics (Eurofins Genomics Europe Shared Services GmbH, Konstanz, Germany).

Assembled 16S rRNA gene sequences were compared against the GenBank database (https://blast.ncbi.nlm.nih.gov/Blast.cgi) (accessed on 30 November 2020). For multiple sequence alignments, the MEGA7 software was used [18]. The sequences obtained during the current study were deposited in the GenBank database (for accession numbers, see Table 1). To compare the relatedness among *P. larvae* ERIC genotypes, complete genome sequences of genotypes I–IV (GenBank accession numbers NZ_CP019651, NZ_CP019652, NZ_CP019655, and NZ_CP019659, respectively) were downloaded from the GenBank database and the average nucleotide identity was calculated using the web-based ANI/AAI-Matrix tool freely available at http://enve-omics.ce.gatech.edu/ani/ (accessed on 15 February 2021) [19].

## 3. Results

A total of nine *P. larvae* strains were included in the study. The *P. larvae* strain CCM 38 (= NCDO 1121 = NRRL NRS-1283 = CCUG 7427) is a type strain of *P. larvae* subsp. *pulvefaciens*; CCM 4484, CCM 4488, and CCM 5680 strains were isolated from the dead larvae obtained from Czech territory; and the rest of the strains (D3, D4, 5S, 10S, and M1) were obtained from infected honeycombs in Slovakia.

The analysis of *P. larvae* isolates by rep-PCR using ERIC primers yielded different fingerprint patterns consisting of bands ranging from approximately 0.25 to 3 kb (Figure 1). In accordance with the proposed genotyping of *P. larvae* [8,9] based on the presence and/or absence of representative bands, three different ERIC genotypes were detected. *P. larvae* strains D3, D4, 5S, 10S, and M1, isolated from beehives in the territory of the Slovak Republic, were identified as ERIC II genotypes. The isolates exhibited typical fingerprint patterns with a specific band around 970 bp and one prominent band migrating at around 2800 bp. Three *P. larvae* strains collected from the Czech Republic (CCM 4484, CCM 4488, and CCM 5680) presented similar banding patterns to the previous ones, but lacked any bands migrating around 2800 bp, thus belonging to the ERIC I genotype. The reference strain CCM 38, the representative of the ERIC IV genotype, was characterized by having a typical band around 500 bp and lacking a 970 bp band.

The results of the sequence comparisons unambiguously classified all isolates as *P. larvae* with sequence identities greater than 99% (data not shown). To study the sequence heterogeneity of 16S rRNA genes among *P. larvae* isolates, multiple sequence alignment was performed. As documented in Table 1, comparative analysis revealed very limited variability among the tested isolates. Only three nucleotide positions (57, 731, and 1022) were observed to be polymorphic (transitions) over the entire 16S rRNA sequence range analyzed.

The MALDI-TOF spectra of the individual bacterial strains were recorded and compared to the mass spectra of the *P. larvae* reference strains, included in the Bruker Daltonics Database. The interpretation of the results was performed according to the manufacturer’s instructions. Score values ≥ 2.0 were accepted for reliable identification at the species level, values between 2.0 and 1.7 indicated probable genus identification, and scores below 1.7 were considered as unreliable identification. The protein mass spectra of all *P. larvae* isolates analyzed presented important signal intensities in the mass range of 2–20 kDa. Seven strains were validly identified as *P. larvae* with a high identification score of over 2.0, and two strains yielded scores of 1.937 and 1. 988 (see Table 1).

A closer inspection of the reference strain DSM 7030T mass spectrum revealed the presence of three major peaks, the dominant one exhibiting a 4293 m/z ratio. Although present in all protein spectra, this peak was not the dominant one in all the studied strains. The second major peak of DSM 7030T was located at 7534 m/z. Again, this peak was identified in all the strains, but in most of them (5S, 10S, CCM 4484, CCM 4488, and CCM 38), its intensity was significantly lower. Interestingly, the isolate CCM 38, which was unambiguously identified as *P. larvae* with a score of 2.081, produced a completely unique protein peak signature with the most dominant peak being presented at 6143 m/z. In summary, although most strains were correctly identified as *P. larvae*, they produced considerably variable MALDI-TOF protein fingerprints, and the dominant peaks differed significantly between the individual strains (Figure 2).

Based on the similarity of the protein profiles obtained by MALDI-TOF analysis, a dendrogram presenting the relatedness of bacterial strains was constructed (Figure 3). Using this approach, the strains included in the study were divided into four biotypes. While all strains of Slovak and Czech origins produced very similar mass spectra and grouped together within closely related MALDI biotypes III and IV, the reference strain CCM 38 (ERIC IV) remained separate and on its own at a distance level of 800. Finally, *P. larvae* reference strains DSM 3615 and DSM 7030T branched from all other isolates at a distance level as high as 1000, forming biotype I.

## 4. Discussion

In recent decades, we have witnessed a dramatic decline in pollinators’ abundance due to various factors. *P. larvae*, the etiological agent of American foulbrood, a widespread and fatal honey bee disease, poses one of the greatest threats to honey bee populations worldwide. Despite being clonally related, *P. larvae* isolates causing outbreaks are not necessarily genetically identical and their subtyping is an essential epidemiological tool in infection prevention and control.

At present, almost exclusively DNA-based methods are used for the genotyping of *P. larvae*. Among them, the enterobacterial repetitive intergenic consensus polymerase chain reaction (ERIC-PCR) was proven to be the most valuable method. Using this approach, *P. larvae* isolates were previously classified into four different genotypes. The genotypes (ERIC I–IV) were shown to correlate with several phenotypic characteristics, including differences in the virulence and severity of the disease [8]. While ERIC I and II are considered to be the most important genotypes, frequently isolated from afflicted hives, ERIC III and IV probably represent historical isolates and only exist in culture collections [4,20,21]. Only recently, the new genotype V was detected among *P. larvae* isolates collected from Spain [9], indicating that more unknown ERIC genotypes may exist in nature.

In our experiments, the majority of *P. larvae* isolates were observed to represent ERIC I and II genotypes based on the banding patterns produced (Figure 1). One of the isolates tested (10S) was classified as an ERIC II genotype, despite its lack of the characteristic 2800 bp band and, instead, presentation of a unique band migrating at approximately 4200 bp. One may speculate that isolate 10S represents a novel, to date undescribed, ERIC genotype. However, further studies are required to evaluate this hypothesis. The CCM 38 isolate, the reference strain for the ERIC IV genotype, produced a clearly distinguishable banding pattern that separated it from the other isolates.

Although DNA-based methods have been widely applied and accepted as standard approaches in the identification and subtyping of microorganisms, they are relatively laborious and time-consuming, thus presenting limited utility in real-time applications. In recent years, techniques based on unique protein signatures have emerged as promising alternatives, especially for the epidemiological study of bacteria. In this context, MALDI-TOF MS offers the advantage of speed and simplicity [22]. The method was used for the identification of different Gram-positive bacteria, including *Paenibacillus* sp. isolates of human origin [23,24] or from plant tissues [25]. To the best of our knowledge, this is the only report on the application of the MALDI-TOF technique for the analysis of *P. larvae* subspecies diversity [26]. The authors used MALDI-TOF mass spectrometry for *P. larvae* isolate discrimination, showing that ERIC I and II genotypes can be unambiguously identified on the basis of mass spectra. For spectra acquisition, an Ultraflextreme mass spectrometer (Bruker Daltonik, Bremen, Germany), providing ultra-high resolution and mass accuracy, is used. However, for routine microbiological diagnostics, two commercially available MALDI-TOF systems, Microflex LT (Bruker Daltonics, Bremen, Germany) and Vitek MS (bioMérieux, Marcy l’Etoile, France), are more frequently used.

In our experiments, a Microflex LT instrument was employed for the identification and typing of fresh *P. larvae* isolates collected from Slovakia and from the culture collection in the Czech Republic. High-quality mass spectra with good reproducibility were acquired, and all isolates were identified as *P. larvae* using a database implemented by Bruker (version 3.3.1.0). However, some isolates yielded intermediate identification scores (values between ≥1.7 and <2), which were only considered as reliable for genus identification.

To confirm the phylogenetic affiliation, the small subunit ribosomal RNA genes of all the isolates included in the study were amplified, sequenced, and compared against the NCBI prokaryotic 16S ribosomal RNA database using the BlastN algorithm. All isolates were unambiguously assigned to the *P. larvae* species, presenting similarities as high as 99.4% to the 16S rRNA sequences of *P. larvae*-type strains (data not shown). Furthermore, a comparative sequence analysis revealed the very limited heterogeneity of 16S rRNA genes, indicating that the *P. larvae* variability observed by ERIC-PCR genome fingerprinting was not accompanied by diversity at the 16S rRNA level. The majority of diversity traits was observed in variable regions V1, V4, and V6 represented by three polymorphic nucleotide positions, and at least seven sequence signatures were detected (Table 1).

No sequence signature associated with either genotype was detected. Generally, all *P. larvae* strains were very similar at the genomic level. The average nucleotide identity between the genome sequences of ERIC-genotype-I–IV representatives was observed to be extremely high (99.6%, data not shown).

To evaluate the discriminatory power of MALDI-TOF MS and ERIC-PCR for the typing of *P. larvae* isolates, a dendrogram based on MALDI-TOF mass spectra MSP (main spectra profile) was generated and compared to ERIC-PCR outcomes. Four distinct MALDI biotypes were detected in the MSP dendrogram at the distance level of 400 or longer. Quite surprisingly, no clear correlation between ERIC genotypes and MALDI biotypes was observed. Two spectra of *P. larvae* are included in the Bruker MALDI database: the spectrum of *P. larvae* DSM 7030—the reference strain for the ERIC I genotype—and the spectrum of *P. larvae* DSM 3615—the reference strain for the ERIC IV genotype [8]. These two strains were clustered separately from the tested isolates and, despite being distinct ERIC genotypes, they divided into branches linked at a very short distance level. Interestingly, the isolate CCM 38, another representative of the ERIC IV genotype, formed a separate clade with an extremely long distance from ERIC IV reference strain *P. larvae* DSM 3615. The remaining isolates, belonging to the ERIC I and II genotypes, were placed in two related MALDI biotypes with distance levels slightly longer than 400. This apparently contradicted the results published by Schäfer et al. [26], who reported that ERIC I and II genotypes could be identified on the basis of mass spectra. In our experiments, the MALDI-TOF MS technique could distinguish ERIC IV from ERIC I and II genotypes, but not ERIC I from II. In a similar study conducted by Schäfer et al. [26], only 10% of ERIC I or II genotype isolates were misidentified using MALDI-TOF MS. These diverging conclusions might have resulted from the different instruments used for the MALDI-TOF analysis (Bruker Ultraflextreme versus Microflex mass spectrometer) or from the varying quality of the biological material. For *Paenibacillus* sp., similar to other Gram-positive sporulating bacteria, several factors, such as culture conditions and degree of sporulation, were shown to affect the quality of mass spectra and identification accuracy [27].

Our data indicate that MALDI-TOF MS reveals much greater biodiversity among *P. larvae* isolates than commonly used DNA-based methods—ERIC-PCR and 16S rRNA gene analysis. For species delineation in the MALDI-TOF analysis, distance levels shorter than 500 have been described as reliable [28]. However, for some bacterial genera, much shorter distance levels were observed between well-defined species, e.g., *Streptomyces* spp. was identified with a cut-off distance value as short as 300 [29]. The diversity observed in MALDI biotypes III and IV could not be attributed to genetic variability at the 16S rRNA level, and further studies are necessary to evaluate its biological significance.

## 5. Conclusions

Our study showed how MALDI-TOF MS can be more useful for epidemiological studies (in our case, a causative agent of AFB—*Paenibacillus larvae*) compared to DNA-based methods (ERIC-PCR and 16S rRNA). MALDI-TOF represents a rapid, inexpensive, and reliable approach for studying the protein profiling of bacteria, which presents a different perspective on the variability of bacteria. The addition of the type-specific reference spectra of *P. larvae* isolates can speed up routine diagnostics. Additionally, further studies are required.

## Figures and Tables

**Figure 1 vetsci-09-00521-f001:**
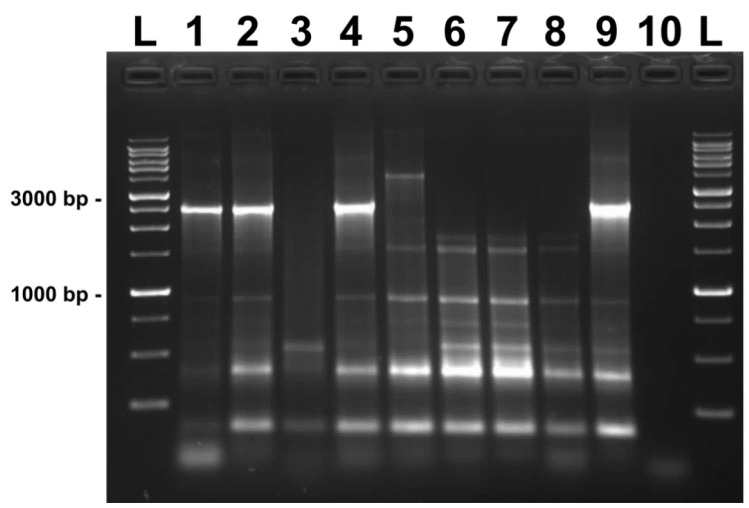
ERIC-PCR fingerprint patterns of *P. larvae* isolates on 1% agarose gel. Lanes: GeneRuler™ 1 kb DNA Ladder; 1: D3; 2: D4; 3: CCM 38; 4: 5S; 5: 10S; 6: CCM 4484; 7: CCM 4488; 8: CCM 5680; 9: M1; 10: negative control. Size of selected marker bands is shown in base pairs.

**Figure 2 vetsci-09-00521-f002:**
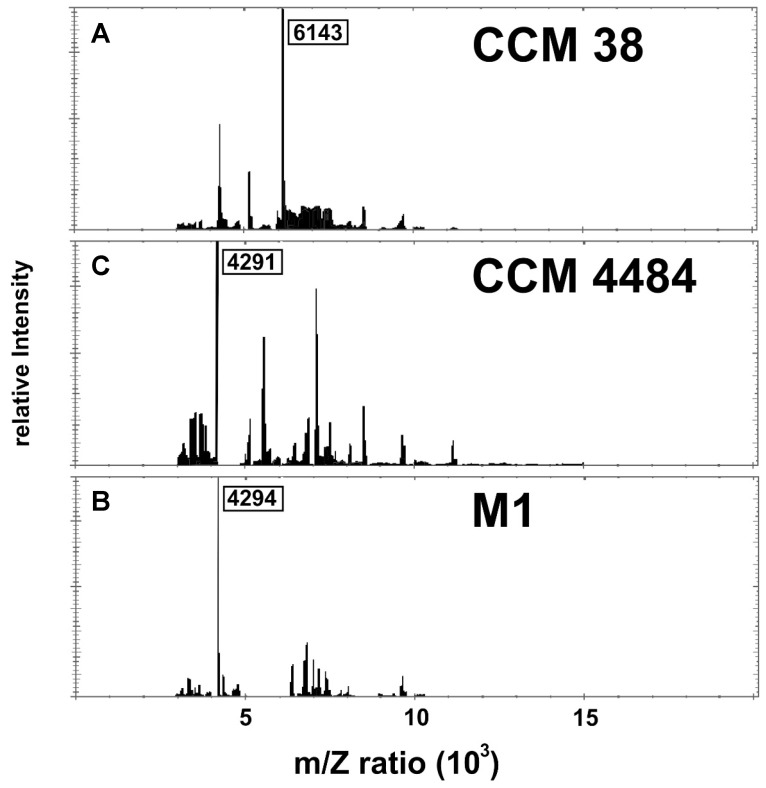
MALDI-TOF spectra of representative *P. larvae* isolates. (**A**) isolate CCM 38: biotype II; (**B**) isolate M1: biotype III; (**C**) isolate CCM 4484: biotype IV. The m/z values of dominant bands are shown.

**Figure 3 vetsci-09-00521-f003:**
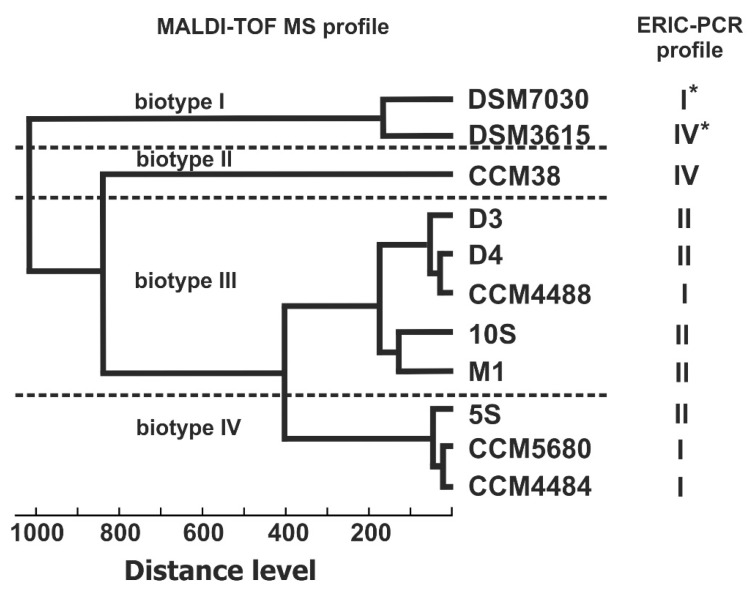
Similarity dendrogram of *P. larvae* isolates based on comparison of MALDI-TOF protein spectra. For DSM7030 and DSM3615, genotype data obtained from the literature [8] and the main spectra profile included in the Bruker database (version 3.3.1.0) were used (marked by asterisk).

**Table 1 vetsci-09-00521-t001:** MALDI-TOF identification and 16S rRNA sequence variability in *P. larvae* strains. Numbering according to *P. larvae* DSM 7030 sequence; NA: not available.

P. larvae Strain	MALDI Identification		16S rRNA GenBank Accession Number	Nucleotide in Variable Positions in 16S rRNA Gene		
	Score	Biotype		57	731	1022
**5S**	2.094	IV	MN613427	T	A	C
**10S**	2.095	III	MN613428	T	A	C
**CCM 4488**	2.014	III	MN613430	C	G	T
**CCM 4484**	1.937	IV	MN613429	C	G	C
**CCM 38**	2.081	II	MN613431	C	A	T
**CCM 5680**	2.072	IV	MN613432	C	G	C
**D3**	2.021	III	MN613433	T	A	C
**D4**	1.988	III	MN982887	C	A	C
**M1**	2.104	III	MN982886	T	A	C
**DSM 7030**	NA	I	AY530294	C	A	T
**DSM 3615**	NA	I	AY530295	C	A	T

## Data Availability

The 16S rRNA sequences obtained during this study were deposited in the GenBank database under accession numbers MN613427-MN613433 and MN982886- MN982887.

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
