# Peer review of "The Application of MALDI-TOF MS for a Variability Study of Paenibacillus larvae"

_vetsci, 2022, doi:10.3390/vetsci9100521_

Round 1
Reviewer 1 Report
Two DNA based technologies and MALDI-TOF have been used to investigate the variability of Paenibacillus larvae. According to the title of the manuscript, the focus is on MALDI-TOF. In many other contexts, use of MALDI-TOF is related to rapid identification of bacteria, or confirmation of presumptive colonies. The approach shows therefore a new application of MALDI-TOF.
For the MALDI-TOF studies in the present work, a full extraction protocol has been used. In routine confirmation studies, colonies are analysed without any extraction or with a simple extraction with formic acid. It is possible, even likely, that the variability observed in the studies in this work would not have been detected with these simpler protocols.
Studies of variability between strains indicate that many strains are included in the study. Here, it appears as 12 strains are included, and data shown for 9 or them. These are few strains, but if few strains are available, it can be justified.
Fingerprinting techniques, a combination of PCR and electrophoresis on 1 % agarose gel, has been applied. A paper print of the manuscript may not give a fair presentation of the gel (Figure 1). However, it is clear that some lanes have few, strong bands, others have weak bands, some have strong bonds around 3000 bp, while others have no bonds in this area at all. Even though similarity of bonds can be seen between lanes, this uneven strength and placement of bonds makes it difficult to judge whether the similarities and differences seen are due to strain variation or rather to different efficiency of the DNA extraction, degrading of template DNA in some cases but not in others, or other reasons. This is a weak point of the study. Even though it strengthen the message that fingerprinting has drawbacks compared to other techniques, it introduces a doubt that the differences seen in the mass spectra from MALDI-TOF is also due to different extraction efficiencies. The result is that the manuscript leaves a doubt of the experimental work in the manuscript in general. This may be unfair, but need to be clarified.
An identification score of 2 or higher for MALDI-TOF analysis was used as cut-off. Two strains gave scores below this value, but according to Table 1, very close to 2.0. Actually, the range of scores is 1.937-2.104, with a median 2.072. This is a small range, lower than expected if the spectra look as different as Figure 2 shows.
The authors describe different peaks for some strains. Instead of questioning the identification of the strains, the authors states that MALDI-TOF is suitable to detect variation. Even though it is possible to follow the thought, the relevance of a method where you first need to know the identity of the strain, and then see if MALDI-TOF show variability, indicate that the methodology have a potential but need further development before it becomes useful in routine analyses. This doubt, and the different qualities of the lanes in Figure 1, indicates that it would be easier to trust the results and assessments if the raw spectra from MALDI-TOF of all spectra were shown.
The manuscript shows that further studies of Paenibacillus larvae is relevant. The present work is a good start, but at the present stage the main message from it is that different methods show different variation patterns between relatively few isolates without further explanation why it is so.
Line 151: “Kostas lab” need a full name and address.
Author Response
Response to Reviewer 1 Comments
Point 1: Two DNA based technologies and MALDI-TOF have been used to investigate the variability of Paenibacillus larvae. According to the title of the manuscript, the focus is on MALDI-TOF. In many other contexts, use of MALDI-TOF is related to rapid identification of bacteria, or confirmation of presumptive colonies. The approach shows therefore a new application of MALDI-TOF.
Response 1: We agree that there are multiple application of MALDI-TOF. Studied set of P. larvae strains was found showed very limited variability at 16S rRNA level but the strains were found to be phenotypically very variable and this variability was observed at MALDI-TOF fingerprints as well.
Point 2: For the MALDI-TOF studies in the present work, a full extraction protocol has been used. In routine confirmation studies, colonies are analysed without any extraction or with a simple extraction with formic acid. It is possible, even likely, that the variability observed in the studies in this work would not have been detected with these simpler protocols.
Response 2: We agree that both protocol and MALDI-TOF apparatus used influence the observed degree of variability - e.g. see differences between our results and Schäfer et al. discussed in Discussion part.
Point 3: Studies of variability between strains indicate that many strains are included in the study. Here, it appears as 12 strains are included, and data shown for 9 or them. These are few strains, but if few strains are available, it can be justified.
Response 3: We agree that the number of tested strains is not very high but even using this set of strain unusually high variability was observed.
Point 4: Fingerprinting techniques, a combination of PCR and electrophoresis on 1 % agarose gel, has been applied. A paper print of the manuscript may not give a fair presentation of the gel (Figure 1). However, it is clear that some lanes have few, strong bands, others have weak bands, some have strong bonds around 3000 bp, while others have no bonds in this area at all. Even though similarity of bonds can be seen between lanes, this uneven strength and placement of bonds makes it difficult to judge whether the similarities and differences seen are due to strain variation or rather to different efficiency of the DNA extraction, degrading of template DNA in some cases but not in others, or other reasons. This is a weak point of the study. Even though it strengthen the message that fingerprinting has drawbacks compared to other techniques, it introduces a doubt that the differences seen in the mass spectra from MALDI-TOF is also due to different extraction efficiencies. The result is that the manuscript leaves a doubt of the experimental work in the manuscript in general. This may be unfair, but need to be clarified.
Response 4: As stated in our manuscript, for typing P. larvae using ERIC-PCR fingerprinting the presence/absence of specific band (https://doi.org/10.1016/j.ijmm.2020.151394) is used instead of comparison of profiles, so band intensities do not play effect. The same procedure for sample preparation was used for all tested strains and similar to data reported elsewhere the extraction protocol better results.
Point 5: An identification score of 2 or higher for MALDI-TOF analysis was used as cut-off. Two strains gave scores below this value, but according to Table 1, very close to 2.0. Actually, the range of scores is 1.937-2.104, with a median 2.072. This is a small range, lower than expected if the spectra look as different as Figure 2 shows.
Response 5: MALDI-TOF analysis of our isolates was several times repeated to confirm unusually big differences in profiles obtained. Despite the differences in the spectra all our isoltes were correctly identified by Bruker database. The algorithm used for spectra comparisons is not completely known but peak positions, intensities and frequencies are taken into account to ensure the highest possible accuracy of identification. We have no explanation for this observation now, especially, if genome comparisons of available P. larvae genomes indicate that all genomes are practically identical as stated in our manuscript.
Point 6: The authors describe different peaks for some strains. Instead of questioning the identification of the strains, the authors states that MALDI-TOF is suitable to detect variation. Even though it is possible to follow the thought, the relevance of a method where you first need to know the identity of the strain, and then see if MALDI-TOF show variability, indicate that the methodology have a potential but need further development before it becomes useful in routine analyses. This doubt, and the different qualities of the lanes in Figure 1, indicates that it would be easier to trust the results and assessments if the raw spectra from MALDI-TOF of all spectra were shown.
Response 6: All our strains (newly isolated from infected beehives) were identified by 16S sequence analysis - just because extremely high differences in MALDI spectra. Despite this differences we have observed very acceptable MALDI identification score and extremely low variability at 16S rRNA sequence level. This indicate that all our strains belongs to P. larvae species and variability seen in MALDI-TOF fingrprints is true picture of P. larvae variability.
Point 7: The manuscript shows that further studies of Paenibacillus larvae is relevant. The present work is a good start, but at the present stage the main message from it is that different methods show different variation patterns between relatively few isolates without further explanation why it is so.
Response 7: We agree that further work is neccesary to understand the variability seen in P. larvae.
Point 8: Line 151: “Kostas lab” need a full name and address.
Response 8: Corrected.

Reviewer 2 Report
This is a well set and interesting research on Paenibacillus larvae strains. My only concern is the use of the database spactra as reference material. Why do the authors not investigate the methodology applied to get the spectra? What was the exact methodology used at building the Biotyper database. Was is the same as used by the authors? This is a key question to answer the difference detected in the ERIC PCR, and the MALDI-TOF results. Try to answer this question, or at least discuss the effect of methodological changes on mass spectra.
One more remark. The manuscript deals the MALDI-TOF spectra as "protein" spectra. But who knows what the extract contains? Thus please discuss more datailed the differences in whole cell, and extract based spectra.
Author Response
Response to Reviewer 2 Comments
Point 1: This is a well set and interesting research on Paenibacillus larvae strains. My only concern is the use of the database spactra as reference material. Why do the authors not investigate the methodology applied to get the spectra? What was the exact methodology used at building the Biotyper database. Was is the same as used by the authors? This is a key question to answer the difference detected in the ERIC PCR, and the MALDI-TOF results. Try to answer this question, or at least discuss the effect of methodological changes on mass spectra.
Response 1: In our experiments the strains originating from former Czechoslovakia were used. Both DSM 7030 and DSM 3615 are very old P. larvae strains of unknown origin, however tehy are still prototype strains of ERIC profiles type I and IV. That was reason why the spectra were included in the study. There are no information available which methodology was used to create Biotyper database but we suppose that extraction methods was used asi t provides more reliable and precise spectra. As P. larvae is sporulating bacterium we suppose that other factors, such as culture conditions and degree of sporulation, affect the quality of mass spectra and identification accuracy see e.g. https://doi.org/10.1038/s41598-017-15808-5 - this effects are disussed in the manuscript.
Point 2: One more remark. The manuscript deals the MALDI-TOF spectra as "protein" spectra. But who knows what the extract contains? Thus please discuss more datailed the differences in whole cell, and extract based spectra.
Response 2: Generally is accepted that in case of bacteria proteins (usually ribosomal proteins) form the dominant peaks seen in MALDI-TOF MS spectra. E.g. in 10.1016/j.mimet.2013.07.021 it was shown that 10 of the 13 species-specific bands correspond to ribosomal proteins - so we prefer to keep term protein spectra in our manuscript.
